# Extreme normalization: approximating full-data batch normalization with single examples

## Abstract

While batch normalization has been successful in speeding up the training of neural networks, it is not well understood. We cast batch normalization as an approximation of the limiting case where the entire dataset is normalized jointly, and explore other ways to approximate the gradient from this limiting case. We demonstrate an approximation that removes the need to keep more than one example in memory at any given time, at the cost of a small factor increase in the training step computation, as well as a fully per-example training procedure, which removes the extra computation at the cost of a small drop in the final model accuracy. We further use our insights to improve batch renormalization for very small minibatches. Unlike previously proposed methods, our normalization does not change the function class of the inference model, and performs well in the absence of identity shortcuts.

## 1 Introduction

Batch normalization (BN) is an often-effective tool in training deep models. It achieves such effectiveness by normalizing the elements of a minibatch jointly, and doing so pervasively throughout the model. When BN is used, it is ensured that there are representations throughout the model where the per-dimension first and second moments (means and variances) are fixed throughout training.

Batch normalization is not well-understood theoretically, and is known to introduce undesired, and often surprising, artifacts into training. Because BN relies on the interaction between examples during training, it causes models to perform poorly when training minibatches are small or contain dependencies between examples. This was shown e.g. in (Ioffe, 2017), which also demonstrated that this can be alleviated by carefully using the moving averages of minibatch statistics during training.

The adverse effects of BN, as well as the engineering complexity needed for large minibatches, have led researchers to tackle the problem of normalization that depends only on individual examples, such as (Ba et al., 2016; Wu & He, 2018). These methods, however, change the architecture used for inference, as the normalization needs to be performed during inference as well. While this may lead to a difference in performance, such interventions do not change only the optimization, but also the hypothesis class. On the other hand, inference-time BN can be folded into the rest of the model weights to yield a regular model (e.g. using only convolutions, bias additions, and ReLU). Hence batch normalization does not change the class of models and can be viewed as solely an optimization device. Approaches that change only the optimization and not the architecture include (Zhang et al., 2019; Brock et al., 2021) which remove the normalization entirely, but depend on identity shortcuts in the model.

In this work, we aim to develop a method for performing per-example normalization in a way that does not modify the inference-time architecture. The resulting method combines per-example gradient computation, the maintenance of moving first and second moments, and an aggregation step that joins the information from multiple examples similar to the way that the gradients with respect to the model parameters are commonly averaged over the minibatch in SGD.

## 2   APPROXIMATING POPULATION NORMALIZATION

Consider a model that involves a normalization step, such as $y_i = x_i - \mathbb{E}\,x$ or $y_i = \frac{x_i}{\sqrt{\mathbb{E}\,x^2}}$. Here, $x_i$ is an activation within a model computed for a particular ($i$th) model input, $y_i$ is the normalized activation, and $\mathbb{E}\cdot$ is the expectation taken over the training data distribution. The examples above achieve a zero mean and unit second moment, respectively. We decouple these two types of normalization in what follows.

Performing a normalization that involves the expectation over the data distribution needs to take into account the fact that the normalizers such as $\mathbb{E}\,x$ and $\mathbb{E}\,x^2$ depend on the distribution of the activation $x$ and therefore depend on the model parameters that affect $x$. For example, if $x$ is an output of a layer in a model, then its value depends on the parameters of this layer as well as of the preceding layers. Stochastic optimization relies on computing a stochastic estimate of the gradient $\frac{\partial L}{\partial W}$ of the loss with respect to weights. Without cross-example interactions, an unbiased estimate is given by the gradient for a randomly sampled example or minibatch. With cross-example normalization, however, this does not hold since we no longer have per-example gradients: the computation for an example is influenced by all other examples. This work aims at obtaining an estimate, albeit not necessarily unbiased, of the overall gradient from a subset of the examples, whose computation requires minimal (ideally none) interaction between examples.

Let us consider a general normalization layer of the form $z_i = f(x_i, S)$, where $S$ is defined as the expectation of some statistic of the activation, $S = \mathbb{E}\,s(x)$. Observe that if the model parameters are fixed and the model is structured as a directed acyclic graph, then the normalizers $S$ can be computed for various layers by considering them in the topological order. However, just computing the normalizers is not enough; as they depend on the model parameters, we need to capture this dependence in the gradient calculation. Batch Normalization (Ioffe & Szegedy, 2015) proposes a solution: approximate each expectation $S$ with its minibatch estimate, through which we can backpropagate during training, using the population averages for inference. However, the fact that no layer other than the input observes the activations computed the same way during training as during inference is a likely culprit for the catastrophic failures of batch normalization, when they happen (Ioffe, 2017). We would like to use the same normalizers during training and inference.

We could approximate the normalizers by computing exponential moving averages (EMAs) throughout training. The update rate of the EMA allows us to trade off between the variance (lower weight given to the last minibatch means lower variance) and staleness of the parameters (higher weight for the last minibatch means the statistics being averaged are computed for more-similar model parameters). The question is now how to backpropagate through the normalizers, once the examples other than the ones in the current minibatch are no longer available. One approach was proposed in (Yao et al., 2020). In another, Batch Renormalization backpropagates using only the current minibatch, while using the EMAs in the forward pass. Specifically, batch renormalization normalizes its inputs (with $x_n$ denoting the value for the $n$th training example) as $\frac{x_n - \mu}{\sigma} StopGradient(\frac{\sigma}{\bar{\sigma}}) + StopGradient(\frac{\mu - \bar{\mu}}{\bar{\sigma}})$ where $\mu$ and $\sigma$ are the mean and standard deviation of $x_{1...N}$ and $\bar{\mu}, \bar{\sigma}$ are their EMAs.

### 2.1   COMPUTATION GRAPHS, GRADIENTS, AND JACOBIANS

To better reason about what we would like the minibatch normalization scheme to do, let us consider the graph representing the computation of the full-batch loss. This is a directed, acyclic graph, whose inputs are the input examples (such as images and labels) and the model weights, and whose output is the batch loss. The other nodes represent intermediate computations, such as hidden layer activations. Each node's value is computed as a function (the node's operation) applied to the node's inputs (nodes from which there is an edge to the given node). Each fixed sequence of operations can be represented with such a graph. Gradient-based learning relies on the computation of gradients in this graph, typically computed by backpropagation. As an alternative formulation, it is easy to show that for any weight node $W$ and loss $L$, the gradient $\frac{\partial L}{\partial W}$ can be computed by considering all directed paths in the graph from $W$ to $L$, computing for each such path the product of the Jacobians corresponding to all edges in that path, and adding the products over all the paths. Backpropagation computes this sum of products efficiently via dynamic programming.

**Batch nodes and shared nodes.** For ease of exposition, and without loss of generality, we will consider a full-batch computation graph applied to a finite dataset $(D_{1...N})$, where the $n$th example $D_n$ could contain, for example, an image and its ground-truth label. We will focus our attention on computation graphs each of whose nodes is either a batch node or a shared node. Batch nodes compute a value for each of the data points, while the shared nodes compute quantities that are aggregated over the dataset. Specifically:

**Batch nodes.** The output of a batch node contains one value per training point. One of the batch nodes is the input node that contains the training data set: $D = (D_1 \ldots D_N)$. Other batch nodes take batch nodes and, possibly, shared nodes as inputs, and compute the outputs such that each output value depends only on the corresponding values from the input batch nodes.

**Shared nodes.** Shared nodes compute values that are shared by the entire dataset. Shared nodes include the model *parameters*. The other type of a shared node is an *reduction node*, which has a batch node as the input and computes the average of the corresponding per-example values. Consider a batch node $Q = (Q_1 \ldots Q_N)$ containing a value for each example. Then a reduction node $S$ would compute $S = \frac{1}{N} \sum_n^N Q_n = \mathbb{E}_n Q_n$. To refer to the batch node $Q$ that is the input to the shared node $S$, we will denote $Q = S^{\text{batch}} = (S_1^{\text{batch}} \ldots S_N^{\text{batch}})$. We are considering the setting in which the model loss $L$ is also a shared node, with the total loss computed as the average of per-example losses.

As an example, if the loss is computed as $L = \mathbb{E}_n |W(x_n - \mu) - y_n|$, where $W$ is the parameter to be learned, $(x_n, y_n)$ are the training inputs and outputs, and $\mu = \mathbb{E}_n x_n$, then the input batch node would have elements $D_n = (x_n, y_n)$, the shared nodes are $L$, $\mu$ and $W$, and the other nodes in the computation graph are batch nodes.

**Graph over shared nodes.** For a computation graph consisting of batch and shared nodes, it is easy to prove (by induction over the graph nodes) that the output of a batch node $Q$ can be written as $Q = (f(D_n, \mathcal{S}))_{n=1...N}$, where $f$ is some function and $\mathcal{S}$ is a subset of the shared nodes in the computation graph. In other words, each element of the batch of values computed by the batch node can be computed from the corresponding input element $D_n$ as well as some of the shared nodes. Constructively, such computation is represented by the subgraph of the computation graph consisting of the nodes and edges that can be reached by walking backwards from the given batch node without crossing shared nodes. It follows that the value of a reduction shared node $S$ can be computed as $S = \mathbb{E}_n S_n^{\text{batch}} = \mathbb{E}_n f(D_n, \mathcal{S})$. We will say that $S$ is S-dependent ("S" is for "shared") on $D$ and on each of $R \in \mathcal{S}$. A shared node $S$ is S-dependent on $R$ if $R$ is either a shared node or the input batch node, and there is a directed path from $R$ to $S$ in the computation graph that does not contain intermediate shared nodes.

We can represent the computation of $L$ (which is a reduction node) as a graph whose nodes are the shared nodes and the input batch node, and which has an edge $R \to S$ if in the original computation graph $S$ is S-dependent on $R$. We will refer to this graph as the *S-graph*, and to the original computation graph as the base graph. See Figure 1. Each node in the S-graph performs an operation on its inputs that is represented with a subgraph of the original computation graph. Computing the values of all the nodes in the S-graph is no more expensive than a single forward pass in the original computation graph. Similarly, computing the gradient of a linear combination of nodes in the S-graph, with respect to the model parameters, has the same cost as a single backpropagation in the original computation graph.

**Jacobians between shared nodes.** Consider an edge $S \to T$ in the S-graph, where $S$ and $T$ are shared nodes and $T = f(D, \mathcal{S})$ (for some subset of the shared nodes $\mathcal{S} \ni S$ and the input batch $D$). We define the Jacobian $J(S, T) = \frac{\partial f}{\partial S} \in \mathbb{R}^{|T| \times |S|}$. From the chain rule, we have the gradient of the loss with respect to a parameter $W$: $\frac{\partial L}{\partial W} = \sum_{k, (S_0 \ldots S_k)} J(S_{k-1}, S_k) J(S_{k-2}, S_{k-1}) \ldots J(S_0, S_1)$ where the summation is over all directed paths $(S_0 = L \ldots S_k = L)$ from $W$ to $L$ in the S-graph.

Because the target $T$ of an edge $S \to T$ is a reduction node, we have, $T = \mathbb{E}_n f(D_n, \mathcal{S})$, thus $J(S, T) = \mathbb{E}_n \frac{\partial f(D_n, \mathcal{S})}{\partial S}$. Therefore, the single-example Jacobian $J_n(S, T) = \frac{\partial f(D_n, \mathcal{S})}{\partial S}$ is an unbiased estimate of $J(S, T)$, and can be computed from only the $n$th training example and the values of some shared nodes. However, a product of such unbiased estimates will not give an unbiased estimate of the product of the Jacobians. For example, consider an S-graph with nodes $W, S, L$, edges $W \to S$, $W \to L$, $S \to L$. Then $\frac{\partial L}{\partial W} = J(W, L) + J(S, L) J(W, S)$, but the

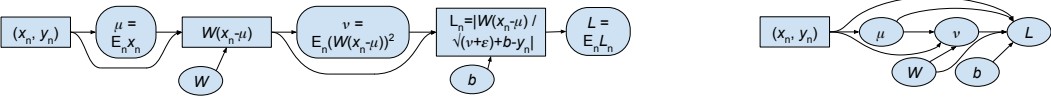

Figure 1: An example computation graph (left) and the corresponding S-graph (right). Shared nodes are ovals, batch nodes are rectangles. The S-graph contains the shared nodes and the input batch node, with an edge between two nodes if a path exists between them in the computation graph that does not pass through other shared nodes. A node in the S-graph computes $\mathbb{E}_n \, f(D_n, \mathcal{S})$ where $(D_n)$ are the training examples and $\mathcal{S}$ is a subset of the shared nodes. An edge $S \to T$ in the S-graph corresponds to the Jacobian $J(S, T)$, and the desired gradient of the loss with respect to model parameters is a sum of products of such Jacobians. This work aims to construct better estimates of such sum of products using a minibatch of examples, or a single example.

approximation of this gradient using a single example (or a minibatch) is not in general unbiased:
$\mathbb{E}_n \, J_n(W, L) + J_n(S, L) J_n(W, S) = J(W, L) + \mathbb{E}_n \, J_n(S, L) J_n(W, S) \neq \frac{\partial L}{\partial W}$.

## 2.2 DEPENDENCIES BETWEEN JACOBIANS LEAD TO BIASED GRADIENTS

Below, we note some of the ways that dependence between the Jacobian estimates in the computation graph manifests itself, and some ways to reduce these dependencies.

**Scaling the second moment gradients.** Consider the divisive normalization $y_n = \frac{x_n}{\sqrt{S}}$, where $x_n$ is the activation in the $n$th example, and $S$ is the second moment of the inputs. For shared nodes $R$ and $T$ with S-graph edges $R \to S \to T$, the single-example estimates $J_n(R, S)$ and $J_n(S, T)$ depend on the corresponding $x_n$ and therefore on each other, resulting in $J_n(S, T) J_n(R, S)$ being a biased estimate of $J(S, T) J(R, S)$. To ameliorate this, we will modify the estimates of the form $J_n(R, S)$ to reduce their dependence on the $n$th example. Observe that if $x_n$ is an output of a linear layer $x_n = W \mathbf{u}_n$ with weights $W$, then both $Q_n = x_n^2$ and $\frac{\partial Q_n}{\partial W}$ scale quadratically with the scaling of $\mathbf{u}_n$. Based on this, we propose scaling $J_n(W, S)$ by $\frac{S}{Q_n}$, such that it is independent of the scale of $\mathbf{u}_n$ (and the expected inverse scaling amount of scaling is 1). We apply the same scaling to $J_n(R, S)$ for all R, which amounts to scaling the gradient as it backpropagates through $S$. If a minibatch of $B$ examples are allowed to interact, and the Jacobians are estimated as $\mathbb{E}_b^B \, J_b(R, S)$, then we scale the gradient by $\frac{S}{\mathbb{E}_b^B Q_b}$. In both cases, the amount of scaling is capped by $f_{\max} = 2$ (that value is not critical).

**Reducing dependence due to per-node normalization and nonlinearity.** Another source of dependence between $J(R, S)$ and $J(S, T)$, where $S$ is the second moment, is the normalization followed by a nonlinearity. For simplicity but without loss of generality, consider a ReLU. As above, the single-example estimate $J_n(R, S)$ depends on $x_n$, even with the gradient scaling above. So does the single-example estimate of $J_n(S, T)$, which has $\frac{\partial ReLU(\gamma y_n + \beta)}{\partial y_n}$ as a factor, which is 0 whenever $\frac{\gamma}{\sqrt{S}} x_n + \beta < 0$. We alleviate that dependence by reducing the degree to which a particular nonlinearity input is affected by any normalized activation. To do so, consider an entire vector of node values $\mathbf{x}_n$, e.g. the output of a neural network layer for the $n$th example. Instead of applying both the normalization and the nonlinearity per node, we will compute $ReLU(\mathbf{R} \cdot (\boldsymbol{\gamma}\mathbf{y}) + \boldsymbol{\beta})$ for some fixed matrix $\mathbf{R}$, with $\cdot$ denoting matrix product. We can set $\mathbf{R}$ to a random and fixed rotation matrix, preserving the total energy in both the forward and back pass. Note that if $\mathbf{x}$ is computed by a linear layer (or convolution), then the rotation by $\mathbf{R}$ can be fused into that linear layer and preserves the inference model class. For some matrix sizes, we can also set $\mathbf{R}$ to a Hadamard matrix, which has the smallest maximum absolute value of the elements. In the convolutional case, we apply a $1 \times 1$ convolution between the normalization and the nonlinearity, multiplying the Hadamard matrix (or a random rotation) by the stack of channel values at each spatial location.

**Shrinking the gradients.** An additional modification to the gradients that we have found very useful is shrinkage. The intuition is that the variance in the estimation of $J(R, S)$ can be reduced by multiplying the estimate by a scalar $r < 1$, and this may in turn reduce the estimation error by finding a more favorable bias-variance tradeoff. When scaling the gradients, we apply the same scaling to $J(R, S)$ for all $R$, and therefore can accomplish this by simply modifying the backpropagation

through $S$. When processing the $n$th example, we replace $S$ with $ScaleGrad(S, S_n^{\text{batch}}, r)$ which returns $S$ but scales the gradients in backpropagation.

**Averaging Jacobians over the minibatch.** Consider a minibatch of $B$ examples, from which we want to estimate the gradient $\frac{\partial L}{\partial w}$, which in turn is the sum of the Jacobian products of the form $J(S_{k-1}, S_k) \ldots J(S_0, S_1)$ (with $S_0 = W$, $S_k = L$). Each of Jacobians could be estimated as the minibatch average. Although these estimates are not independent, this can be effective in practice when combined with gradient scaling, and is what Batch Renormalization does. However, computing the minibatch estimates of all of the Jacobians forces interactions among the examples in the minibatch at each layer during backpropagation. We could estimate all of the Jacobians based on individual examples to avoid the interactions, but this increases the bias in the gradient estimation due to the dependence between the Jacobians. As a compromise between these two extremes, we break the product $J(S_{K-1}, S_K) \ldots J(S_0, S_1)$ into subproducts, such that each of these factors is estimated as the product of per-example estimates of the constituent Jacobians for each example in the minibatch, averaged over the minibatch. A decomposition that both gives good empirical performance and lends itself to efficient computation is into two groups, one of which is the single Jacobian $J(S_{k-1}, L)$ and the other containing all the others:

$$J(S_{k-1}, L) \ldots J(S_0, S_1) \overset{\text{biased}}{\approx} \frac{\sum_b^B J_b(S_{k-1}, L)}{B} \frac{\sum_b^B J_b(S_{k-2}, S_{k-1}) \ldots J_b(S_0, S_1)}{B}. \quad (1)$$

The first factor is the minibatch estimate of the partial derivatives of the $L$ node in the S-graph. It can be computed via backpropagation, as the minibatch average of per-example gradients with respect to the shared nodes, without backpropagating through the shared nodes. The second factor is the biased estimate of the gradient of the linear combination of the shared nodes with respect to the model weights. The product of the two factors is computed by backpropagation from the shared nodes, feeding in the first decomposition factor, and averaging the resulting per-example gradients. The estimate in Eq. (1) is biased even as $B \to \infty$ because e.g. $\mathbb{E}_n J_b(S_1, S_2) J_b(S_0, S_1) \neq J(S_1, S_2) J(S_0, S_1)$. To alleviate the bias, we need to use the rescaling of the gradients through the second moments, and may need to apply the gradient shrinkage, as described above.

## 3 TWO-STAGE BACKPROPAGATION

We present a complete description of the two-stage backpropagation procedure here. We consider a model that contains the subtraction of the mean and division by the square root of the second moment. We estimate the gradients using a minibatch of size $B$, use the estimates (such as exponentially moving averages, EMAs) of the shared nodes in the forward pass, and modify the backpropagation of the gradients.

### 3.1 MODIFYING THE NORMALIZATION

For generality we consider the convolutional case where multiple values in the model share the normalization statistics. Let $\mathbf{x}_b = (x_{b,1\ldots d})$ be the vector being normalized corresponding to the $b$th example with $d$ being the number of units normalized jointly, such as the number of spatial locations in the convolutional map. Below we show the computation of the subtractive normalization that subtracts the data mean $\mu$, and divisive normalization that divides by the square root of the data second moment $\nu$. Let us define $ScaleGrad(S, Q, r)$ for a scalar $r$ as the operation returning its first argument $S$, but propagating the gradient to $Q$ after scaling it by $r$, so that $ScaleGrad(S, Q, r) = StopGradient(r)Q + StopGradient(S - rQ)$.

**Subtractive:** Define per-example mean $\mu_b = \mathbb{E}_i^d x_{b,i}$, and current population mean $\overline{\mu} = \text{EMA}(\mu_b)$. Define $\mu_b^\dagger = ScaleGrad(\overline{\mu}, \mu_b, r_\text{m})$ for gradient shrinkage $r_\text{m}$ (1 by default). Output $(\mathbf{x}_b - \mu_b^\dagger)$.

**Divisive:** Let $\nu_b = \mathbb{E}_i^d x_{b,i}^2$, $\overline{\nu} = \text{EMA}(\nu_b)$, $\nu_b^\dagger = ScaleGrad(\overline{\nu}, \nu_b, r_\text{v} \min(\frac{\overline{\nu}}{\nu_b}, f_\text{max}))$. Output $(\frac{\mathbf{x}_b}{\sqrt{\nu_b^\dagger + \epsilon}})$.

Such normalizations do not involve example interactions in backpropagation. If the examples in the minibatch are allowed to interact (as in Batch Renormalization), the subtractive normalization would be modified to use $\mu = \mathbb{E}_b^B \mu_b$ instead of $\mu_b$, and the divisive normalization to use $\nu = \mathbb{E}_b^B \nu_b$ instead of $\nu_b$.

### 3.2 MODIFYING THE BACKPROPAGATION

We use the estimates from Eq. (1) by breaking the backpropagation into two parts, interleaved with gradient aggregation. Let $L_b$ is the loss computed on $b$th example in the minibatch, with the minibatch loss defined as $\ell = \mathbb{E}_b^B L_b$. We will refer to the set of the means in the model as $\{\mu\}$, the set of the second moments as $\{\nu\}$, and the weights $\{W\}$.

In Sec. 3.1 we modified the normalization such that *backpropagation does not require inter-example interactions*. Now let us add extra nodes that cause the per-example gradients to interact. For the subtractive normalization, let $\mu^\ddagger = \textit{ScaleGrad}(\overline{\mu}, \mathbb{E}_b^B \mu_b, r_{\mathrm{m}})$. For the divisive normalization, let $\nu^\ddagger = \textit{ScaleGrad}(\overline{\nu}, \mathbb{E}_b^B \nu_b, r_{\mathrm{v}} \min(\frac{\overline{\nu}}{\mathbb{E}_b^B \nu_b}, f_{\max}))$. Note that the gradients of $\mu^\ddagger$ and $\nu^\ddagger$ can be computed by combining per-example gradients computed separately from each other.

**(a)** Backpropagate from the minibatch loss $\ell$ to all $\{\mu_b^\dagger\}$, $\{\nu_b^\dagger\}$, and $\{W\}$, without backpropagating through any of those nodes. This can be done for each example independently. Denote the resulting gradients $\{g_b^\mu\}$, $\{g_b^\nu\}$ and $\{g^W\}$, respectively. Define $g^\mu = \sum_b^B g_b^\mu$, $g^\nu = \sum_b^B g_b^\nu$. These are estimates of Jacobians of the form $J(S, L)$.

**(b)** Feed in the gradients from the previous step into backpropagation starting from $\{\mu^\ddagger\}$ and $\{\nu^\ddagger\}$, i.e. compute the gradients, with respect to $\{W\}$, of $\sum_{\mu \in \{\mu\}} g^\mu \mu^\ddagger + \sum_{\nu \in \{\nu\}} g^\nu \nu^\ddagger$ where $g^\cdot$ are considered constant. This can be done for each example separately followed by adding the gradients. Denote the gradients from this step as $\{h^W\}$.

**(c)** The total gradients are $\{g^W + h^W\}$.

Compared to standard backpropagation in a model with batch normalization (or batch renormalization), the above procedure uses roughly twice the amount of computation because the backward pass is performed twice. What is gained is the ability to process one example at a time.

### 3.3 NORMALIZING CONVOLUTIONAL MODELS

We compute the mean before the linear operation and the second moment after it, which removes the dependence of the mean on the weights of the given layer. With batch normalization, subtracting the mean before or after the linear operation is equivalent, but in our case they are not due to the staleness of the moving averages; computation of mean $\boldsymbol{\mu}$ before the linear transform removes the dependence of the EMA estimate of the mean on the layer weights $\mathbf{W}$. If the inference architecture is to add the bias after the linear operation and not before, we equivalently replace $\mathbf{W}(\mathbf{x} - \boldsymbol{\mu}) = \mathbf{W}\mathbf{x} - \mathbf{W}\boldsymbol{\mu}$. In convolutional models with circular padding, we can similarly replace $\mathbf{W} \star (\mathbf{x} - \boldsymbol{\mu})$ with $\mathbf{W} \star \mathbf{x} - \widetilde{\mathbf{W}}\boldsymbol{\mu}$ where $\widetilde{\mathbf{W}}$ is the sum of the convolutional kernel $\mathbf{W}$ along the spatial dimensions. We apply the same modification even with other padding types; in such cases subtracting $\widetilde{\mathbf{W}}\boldsymbol{\mu}$ does not make the normalized activations zero-mean, but we have not found this to be an issue in practice.

### 3.4 TWO TOWERS

The combination of the normalization modification (to avoid gradient aggregation) and the two-step gradient computation has another interpretation, in which two model towers are created. The first tower uses the modified normalization and also computes the aggregated moments $\{\mu^\ddagger\}$ and $\{\nu^\ddagger\}$. Those are used to perform the normalization in the second tower, and the minibatch loss is set to the loss computed from the second tower. In backpropagation, gradients for different examples interact only when adding the gradients with respects to the weights, and when backpropagating through $\{\mu^\ddagger\}$ and $\{\nu^\ddagger\}$. Therefore the gradient computation can be broken up into two stages, the first backpropagating (in the second tower) to $\{\mu^\ddagger\}$ and $\{\nu^\ddagger\}$ and the second continuing from there (in the first tower), which amounts to two sets of per-example computations, with the gradient aggregation after each of the two sets.

We show the construction of a convolutional layer in such two towers in Alg. 1. There the layer receives the minibatches $\mathbf{x}$ in the first tower and $\mathbf{x}'$ in the second. The two sets of values, as well as the outputs of the corresponding layers in the two towers, are equal in the forward pass, but the gradients propagate differently between the towers. We also show how the corresponding layer is constructed in the inference model, such that at the inference time it acts like a standard convolution followed by bias and the ReLU. One additional modification the algorithm introduces is scaling the

**Trainable:** $\mathbf{W}$ (size $H \times H \times C_i \times C_o$, for convolution spatial support $H \times H$); $\boldsymbol{\beta}, \boldsymbol{\gamma}$ (size $C_o$)
**Const:** scalars $r_m, r_v, f_{max}, \epsilon; u_{max}$ (bound on second moment of normalized activations during training); rotation $\mathbf{R}$ (size $C_o \times C_o$; optional)
**Moving averages:** $\overline{\boldsymbol{\mu}}$ (size $C_i$), $\overline{\boldsymbol{\nu}}$ (size $C_o$)

**Function** *ConvLayer_Train*$(\mathbf{x}, \mathbf{x}')$:    $\triangleright$ Input sizes: $B \times S \times S \times C_i$ (for spatial size $S \times S$)
  $\boldsymbol{\mu} = \mathbb{E}_{i,j} \mathbf{x}_{\cdot,i,j,\cdot};$   $\boldsymbol{\mu}^\dagger = ScaleGrad(\overline{\boldsymbol{\mu}}, \boldsymbol{\mu}, r_m)$   $\triangleright$ size $B \times C_i$
  $\mathbf{p} = conv(\mathbf{x}, \mathbf{W}) - \boldsymbol{\mu}^\dagger(\sum_{i,j} \mathbf{W}_{i,j,\cdot,\cdot})$
  $\boldsymbol{\nu} = \mathbb{E}_{i,j} \mathbf{p}_{\cdot,i,j,\cdot}^2 + \epsilon;$   $\boldsymbol{\nu}^\dagger = ScaleGrad(\overline{\boldsymbol{\nu}}, \boldsymbol{\nu}, r_v \min(\frac{\overline{\boldsymbol{\nu}}}{\boldsymbol{\nu}}, f_{max}))$
  $\mathbf{q} = \frac{\mathbf{p}}{\sqrt{\boldsymbol{\nu}^\dagger}} StopGradient(\min(1, u_{max}\sqrt{\frac{\overline{\boldsymbol{\nu}}}{\boldsymbol{\nu}}}))$
  $\boldsymbol{\mu}^\ddagger = ScaleGrad(\overline{\boldsymbol{\mu}}, \mathbb{E}_b \boldsymbol{\mu}_{b,\cdot}, r_m);$   $\boldsymbol{\nu}^\ddagger = ScaleGrad(\overline{\boldsymbol{\nu}}, \mathbb{E}_b \boldsymbol{\nu}_{b,\cdot}, r_v \min(\frac{\overline{\boldsymbol{\nu}}}{\mathbb{E}_b \boldsymbol{\nu}_{b,\cdot}}, f_{max}))$
  $\mathbf{q}' = \frac{conv(\mathbf{x}', \mathbf{W}) - \boldsymbol{\mu}^\ddagger(\sum_{i,j} \mathbf{W}_{i,j,\cdot,\cdot})}{\sqrt{\boldsymbol{\nu}^\ddagger}} StopGradient(\min(1, u_{max}\sqrt{\frac{\overline{\boldsymbol{\nu}}}{\boldsymbol{\nu}}}))$
  *UpdateEMA*$(\overline{\boldsymbol{\mu}}, \mathbb{E}_b \boldsymbol{\mu}_{b,\cdot});$   *UpdateEMA*$(\overline{\boldsymbol{\nu}}, \mathbb{E}_b \boldsymbol{\nu}_{b,\cdot})$
  **return** $ReLU(Rotate(\mathbf{q} \odot \boldsymbol{\gamma}) + \boldsymbol{\beta}), ReLU(Rotate(\mathbf{q}' \odot \boldsymbol{\gamma}) + \boldsymbol{\beta})$

**Function** *ConvLayer_Inference*$(\mathbf{x})$:
  $\mathbf{K} = \frac{\mathbf{W} \odot \boldsymbol{\gamma}}{\sqrt{\overline{\boldsymbol{\nu}}}}\mathbf{R};$   **return** $ReLU(conv(\mathbf{x}, \mathbf{K}) + (-matmul(\overline{\boldsymbol{\mu}}, \sum_{i,j} \mathbf{K}_{i,j,\cdot,\cdot}) + \boldsymbol{\beta}))$

**Function** *Rotate*$(\mathbf{y})$:   **return** $conv(\mathbf{y}, reshape(\mathbf{R}, 1 \times 1 \times C_o \times C_o]))$

**Algorithm 1:** Convolutional layer for training and inference. The training inputs are minibatches $\mathbf{x}$ and $\mathbf{x}'$ which have the same forward values but differently defined gradients. We build two towers, where the first has the gradients defined to enable per-example gradient computation, and computes the moments used to perform the normalization in the second tower. The loss used for backpropagation is computed for the second tower. The operator *ScaleGrad*$(S, Q, r)$ behaves like *StopGradient*$(r)Q$ in backpropagation, but returns $S$ in the forward pass. The training loss is computed from the second tower, corresponding to $\mathbf{x}'$.

activations to prevent their per-example norm from becoming too large, which happens rarely but can cause instability in training. No such scaling is applied at the inference time.

## 4 RELATED WORK

There has been a substantial body of work attempting to reduce or remove the minibatch dependence that is required by batch normalization. One class of approaches changes the architecture such that the normalization is confined to a single example, and the inference architecture contains the same normalization. This includes group normalization (Wu & He, 2018) and layer normalization (Ba et al., 2016), among many others, that normalize sets of activations for an example relative to each other, rather than normalizing the values of the same activation across a set of examples. Such approaches are applicable in situations that batch normalization is not, such as recurrent models. The normalization affects the model class, which can be a disadvantage, as it conflates two distinct properties of the model architecture – on the one hand, being able to capture the function of interest and containing the appropriate inductive biases, and on the other hand being amenable to gradient optimization. This differs from batch normalization, which does not change the function class.

Another class of approaches eliminates the normalization altogether, via a combination of architectural building blocks and appropriate initialization. An example of this is the fixup initialization (Zhang et al., 2019), combining the residual connections (He et al., 2015) with the initialization that causes the residual blocks to act as identity at initialization. This is tightly coupled to the choice of the function class. We aim to provide a method that, like batch normalization, does not change the function class, does not rely on skip connections, and enables training that behaves similarly (in terms of accuracy as the function of the training steps) as batch normalization.

Yet another class of approaches, which ours is closest to, aims to recover the properties of batch normalization while reducing the dependence on the minibatch size. One such work is (Yao et al., 2020), which approximates the gradients via the normalization statistics using the information

collected from multiple examples, while processing only one example at a time – and, in addition, enables the use of that information even when the model has been updated since the other examples have been seen. The experiments in that work have been conducted using ResNets (He et al., 2015), while in this work we deliberately avoid the models with skip connections to avoid dependence on the advantages they provide.

The approach taken by (Chiley et al., 2019) also aims at using single examples, combined with the moving average history of previous example statistics, to approximate the behavior of batchnorm. However, they include an extra step of normalizing the activations in an example, relative to the other activations, changing the function class, which we aim to avoid.

## 5 EXPERIMENTS

We train the Inception-v3 (Szegedy et al., 2015) model (without the side head) on the ImageNet ILSVRC-2012 dataset (Russakovsky et al., 2015). The learning rate for the final classification layer is reduced by $10\times$ to account for the potentially larger activations without the minibatch normalization. The training uses 50 workers, each processing minibatches of size 32 ($32 \times 50 = 1600$ examples per training step). After each step the moving averages are updated at the rate of 0.2. The moving averages are initialized by using the first 200 steps to only update the EMAs but not the weights. For 50 updates after that, each weight update is followed by 50 steps of only updaing the EMAs, to avoid their staleness very early in training when the weights change rapidly (an alternative is to ramp up the learning rate (You et al., 2017)). During training, we clip the per-example second moment of the normalized activations by $u_{\max} = 5$, and cap the scaling of the gradient through second moments by $f_{\max} = 2$ (these values are not critical). Gradient shrinkage is not used ($r_{\mathrm{m}} = r_{\mathrm{v}} = 1$) unless noted. The test accuracy is increased slightly by decaying learning rate at half the speed for the biases as compared to convolution weights, injecting multiplicative (uniform from $[0.9, 1.1]$) and additive (uniform from $[-0.1, 0.1]$) noise after normalization, and setting the dropout keep probability to 0.6.

For the experiments below, we provide the peak validation accuracy along with the number of training steps required to achieve it, as well as the accuracy achieved after the same number of training steps as it takes the baseline model to achieve its peak accuracy.

**Baseline.** We train the Inception model on ImageNet, using batch normalization. The model reaches the peak validation accuracy of **78.5% at 120k steps**. For the experiments that follow, we report the validation accuracy at 120k steps, as well as the peak validation accuracy.

**Two-stage gradient computation.** We replaced batch normalization with the method in sec. 3, which allows the gradients to be estimated processing each example separately, followed by the aggregation of gradients, followed by the second stage of gradient computation processing each example separately and using the above aggregation results. The model accuracy is **77.8% at 120k steps**, increasing to **78.2% at 150k steps**.

**Different minibatches for two stages.** While the method in sec. 3 replicates a minibatch, the minibatches used for the two parts of gradient computation do not need to be the same. We have found that the only a small reduction in the model performance results from using a much smaller minibatch for the second stage of the gradient computation (backpropagating from the shared nodes other than the loss). We augmented a minibatch of 32 examples with an extra minibatch of 4 examples, for the resulting minibatch $(\mathbf{x}_{1:4}; \mathbf{x}'_{1:32})$. All $4 + 32$ examples are sampled from the training dataset, but the labels for $\mathbf{x}_{1:4}$ are not used. This gives the accuracy of **77% at 120k steps**, which increases to **77.6% at 150k steps**.

**Improving batch renormalization.** Batch renormalization (Ioffe, 2017) allows the use of smaller minibatches for normalization, compared to BN. Here, we show how to improve it. Instead of computing the gradient in two passes as in sec. 3, we can perform cross-example normalization at every layer. To enable this, we use the statistics $\mu^{\ddagger}$ and $\nu^{\ddagger}$ instead of $\mu^{\dagger}$ and $\nu^{\dagger}$. This removes the need for two minibatches and the doubling of computation, but causes the example gradients to interact at every layer.

We use small microbatches (of size 2 or 4) for cross-normalization, with the gradients from multiple microbatches (total 1600 examples) added to perform an update. When normalizing over microbatches of size 4, we get the accuracy of **77.4% at 170k steps (76.2% at 120k)**. For this microbatch size,

(Ioffe, 2017) reports the smaller peak accuracy of 76.5% (at 130k steps, at which point we get 76.7%). For a microbatch of 2, we obtain the accuracy of **72% at 180k steps (70.5% at 120k)**. This can be improved by using the Hadamard rotation between the normalization and ReLU as described in sec. 2.2, raising the accuracy to **74.8% at 180k steps (73.5% at 120k)**. The rotation and the preceding normalization can be fused for inference with the convolution before them, thus the inference architecture remains unchanged.

Further improvements to the accuracy at small microbatches can be achieved by shrinking the gradients through the means by $r_{\mathrm{m}}$ and through the second moments by $r_{\mathrm{v}}$. For a microbatch of size 2, with Hadamard rotation and $r_{\mathrm{m}} = r_{\mathrm{v}} = 0.5$ we get **77.8% at 155k steps (77.2% at 120k)**.

**Single examples.**  Finally, to push the algorithm to its extreme, we considered the true per-example gradient computation, with no interaction between examples. This is the limiting case of our Batch Renormalization modification, with the microbatch size of 1 (which the original batch renormalization does not allow). Without gradient shrinkage, we trained the model both with and without the Hadamard rotation, with the rotation increasing the validation accuracy by about 5%: 68.3% vs. 63.8% (65.8% vs. 60% at 120k), with vs. without the rotation, respectively. The rotation is critical in this setting, and the performance can be improved further via gradient shrinkage. With a coarse search for $r_{\mathrm{m}}, r_{\mathrm{v}}$, for $r_{\mathrm{m}} = 0, r_{\mathrm{v}} = 0.8$ we achieve the validation accuracy of **75% at 170k steps (73.7% at 120k)**. Although at 120k steps this model performs 5% worse than the baseline model (with BN), our method requires no interaction between the examples, and has the same computation complexity during training as the batch-normalized model.

**Residual networks.**  In addition to the experiments with the Inception architecture, we evaluated the performance of our method on the Resnet-50 architecture (He et al., 2015). We applied batch normalization to each convolutional layer, and also to the linear skip-connections that change the number of channels. The baseline achieved **76.3%** top-1 accuracy. Two-stage gradient computation gave a similar **76.6%**. The improved batch renormalization (described above), normalizing over microbatches of 2 examples, gave **75%** when using the gradient scaling factors $r_{\mathrm{m}} = 0, r_{\mathrm{v}} = 0.8$ and Hadamard rotation.

# 6    DISCUSSION

We have presented an interpretation of batch normalization as a method for approximating population normalization – similar to the way that stochastic gradient computation aims to approximate the full gradients. We seek other solutions with more favorable properties: making the training model and inference model equivalent in the forward pass, and reducing the cross-example interaction in the gradient computation. We propose a method for estimating the gradients that, by reducing this interaction, enables the gradients to be estimated without holding more than one example in memory at any given time, at the cost of doubling the computation, although this can be reduced (at a small cost to the accuracy) by using a smaller minibatch for portions of the gradient estimation. We show how to achieve better performance with batch renormalization for extremely small (2 examples) minibatches, and demonstrate only a modest degradation in performance when avoiding cross-example interactions altogether, wherein the parameter gradient is a simple average of the per-example gradients. Our method does not change the inference architecture of convolutional models, in that the model used for inference is still a convnet without any extra operations such as inference-time normalization. Our experiments are conducted on Inception-v3, which, unlike the state-of-the-art image analysis architectures, lacks skip connections and enables us to show that our proposed approach does not take advantage of the beneficial properties of such shortcuts. We did not conduct experiments on the state-of-the-art architectures that benefit from skip connections.

We proposed several building blocks motivated by reducing the detrimental effects of the bias introduced when the gradients are estimated from minibatches, and products of Jacobians are estimated as products of their stochastic and non-independent samples. This includes scaling the gradients to compensate for the easily-identifiable sources of dependence, introducing the rotation between the normalization and nonlinearity to reduce the gradient dependence, and a scaling applied to gradients as they backpropagate through certain nodes in the computation graph. We suspect that at least some of these building blocks not only enable the reduction in the minibatch size required for normalization, but are applicable to other use cases in which the population gradient is not a mere expectation of per-example gradient over the data, and needs to be estimated from minibatches.

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
