# OpenReview forum: "Extreme normalization: approximating full-data batch normalization with single examples"
_ICLR.cc/2022/Conference — ICLR 2022 Submitted_

### Official Review · Reviewer_bk8j · 2021-10-28

**Correctness:** 3
**Technical Novelty And Significance:** 3
**Empirical Novelty And Significance:** 3
**Recommendation:** 6
**Confidence:** 2

**Main Review:**

The paper includes some interesting insights on normalization. However, I find it difficult to catch the practical insights. The proposed method doesn’t improve the accuracy, but adds computations. Why would a practitioner use it? I understand the idea is to perform online normalization in a manner similar to SGD, but it doesn’t seem to offer any practical advantages over the existing batch-normalization method.

**Summary Of The Paper:**

The paper offers an interpretation of batch normalization as a method for approximating population normalization. The proposed method keeps training model and inference model equivalent in the forward pass, and reduces the cross-example interaction in the gradient computation. Furthermore, the paper introduces a method for estimating the gradients, by reducing interaction between examples, holding a single example in memory at any given time. This comes at the cost of doubling the computation. The computation can be reduced with some cost to the accuracy by using a small mini-batch for parts of gradient estimation.


**Summary Of The Review:**

The motivation for the proposed method is weak. Although the overall insights are interesting. Improving the motivation, especially with regards to the practical aspects (as detailed above) will help me to improve the ranking of the paper.

---

### Official Review · Reviewer_mQAF · 2021-11-01

**Correctness:** 4
**Technical Novelty And Significance:** 2
**Empirical Novelty And Significance:** 2
**Recommendation:** 5
**Confidence:** 4

**Main Review:**

Strength: the idea of altering the computational graph is novel, especially to encompass several samples in a single graph where the weights are 'shared'

Weakness: The computational experiments as I read them show that the new approach is subpar. It seems that the baseline always outperforms. The experiments in general are not presented in a good way. It comes across as 'running out of space' in the experimental section. I suggest to remove or shorten Sections 2 and 3 and improve Section 5. The experiments are also limited to a single dataset.

**Summary Of The Paper:**

In batch normalization (BN) it is common practice to consider statistics over the mini-batch. The authors propose a two step approach for per-sample based normalization that is more accurate (they actually don't prove this) than standard BN. They do this by augmenting the computational graph (in essence the computational graph involves several samples and not just a single sample).

**Summary Of The Review:**

The idea is sound and it should definitely be further studied. The experimental section begs for improvements. Not just the exposition, but the entire study. More datasets should be included. Inception-v3 is a rather old architecture and thus more modern architectures should be considered. ResNet-50 is included which is positive.

Minor remark: The hyper parameters are listed however there is no word how they were selected. Usually with BN gradient clipping is not needed so it is surprising that it is used.
I also do not see a discussion on computational times.

---

### Official Review · Reviewer_CZjS · 2021-11-02

**Correctness:** 2
**Technical Novelty And Significance:** 2
**Empirical Novelty And Significance:** 2
**Recommendation:** 3
**Confidence:** 4

**Main Review:**

**Strengths**

The idea of this work is novel and interesting, I really appreciate it.


**Weakness**

Frankly speaking, I think this submission is more like a technical report rather than a complete paper. I will demonstrate the weakness of it one by one:

1. Missing important literature.

The authors missed a previous work MABN(https://openreview.net/forum?id=SkgGjRVKDS). If you read this paper, you will see MABN also deal with small batch issues of BN without introducing additional operators into the model during the inference time. Similar to this paper's idea, MABN also uses moving average statistics during back propagation to solve the drawback of BRN in an elegant way. MABN brings little extra computation overhead while your method double the times of bp during the training time. In a word, if the proposed method only addresses small batch issues of BN, it's trivial and perform worse than a previous method.

2. The demonstration on intuition in section 2.1, 2.2 is really confusing.

In section 2.1, there are a large amount of text description on some general concepts like computation graph, batch nodes, shared nodes, etc, and the relations among them. I really struggled to get the point why these concepts were needed. As far as I'm concerned, there are only three types of nodes you actually used: weights, feature map(batch nodes), and batch statistics. So why you invent new terms to make readers struggle to understand? Another small complaint is that why you denote jacobian $\frac{\partial T}{\partial S}$ as $J(S, T)$? It should be easier for readers to realize the chain rule if you denote the jacobian $\frac{\partial T}{\partial S}$ as $J(T, S)$.  In a word, I think it will be easier to understand if these concepts can be mathematically formulated. Proper formulas and illustrations is better than verbal descriptions if you aim to propose a general framework;

Besides, I didn't get the main idea of this paper, until the I saw the last sentence in section 2.1. I thought this claim was very interesting, and somehow reasonable, so I looked forward to seeing justification on this claim in the following part. When I saw the title of the section 2.2, the expectation became greater. But what I have read in section 2.2? You began to introduce specific tricks in neural networks consisting of linear, BN, relu layers. Then why you propose a general framework in section 2.1? Why not just directly demonstrate what method you apply on a neural network only consisting of BN, relu, linear(CNN) modules? Besides, I failed to understand why you propose these tricks, the intuition behind these tricks is too vague, I cannot understand the connection between them and the last claim in section 2.1.

3. The experiment result is not convincing.

The most confusing part is why the experiment results are presented in text descriptions instead of charts or tables. I think it will be much easier for readers to read and compare.

Only two experiment setting are examined in this paper: inception-v3 and resent50 on imagenet. And the training setting is really wired: the batch size is $32\times50$. I understand it's perhaps too time-consuming to set a real small batch setting in imagenet experiments, but why you do not use a more standard settings, like $32\times 8$ batch size to verify the effectiveness of the proposed method in inception-v3? Besides, a real small batch cases are needed, like detection task or segmentation task, in which the batch size are usually lower than $8$.

I also notice you claim that " in this work we deliberately avoid the models with skip connections to avoid dependence on
the advantages they provide." Do you imply the previous work (Yao et al. 2020) relies on the skip connections structure? I don't agree. So you need to justify your claim that it is indeed an advantage of your method to work well on non-residual structure, while the counterparts, CBN(Yao et al. 2020) or even MABN, cannot. It means more sophisticated comparison experiments are necessary to make your paper more convincing.

4. Citation format is not correct.

You need to clearly show the source (journal, conference, etc) of these cited papers . Not just to show respect to the previous work, but also help readers to properly assess the reliability of your paper, if you use these cited papers as supportive evidence. Please follow a formal citation style.




**Summary Of The Paper:**

This paper propose a training method to deal with small batch issues of batch normalization. Their proposed method does not need to introduce any extra operators into the model during the inference time. The authors verified the effectiveness of the method in Inception-v3 trained on imagenet.

**Summary Of The Review:**

I like the idea of this work, but obviously the authors do not elaborate or justify their idea in either theoretical or empirical aspects. The results is not significant comparing with existing work, either. Besides, the submission seems not a complement paper due to its writing. I lean to reject it.

---

### Official Review · Reviewer_cEEb · 2021-11-04

**Correctness:** 3
**Technical Novelty And Significance:** 2
**Empirical Novelty And Significance:** 2
**Recommendation:** 3
**Confidence:** 3

**Main Review:**

While the problem of improving BN's performance on small training batch sizes is important, I find that this paper can be significantly improved in several aspects.

1. The approach is rather ad hoc and not properly motivated. As far as I understand, the forward pass of the proposed method is very similar to batch renorm. The main contribution is a series of techniques to modify the back propagation. This makes it both hard to understand as well difficult to adopt in practice.

2. Empirical evaluation is underwhelming. The majority of the experiments are conducted on Inception V3, with the only two baselines being BN and batch renorm. Also, on large batch sizes, the proposed method underperforms BN despite the additional complexity. On small batch sizes, it marginally outperforms batch renorm, but given the scope of the experiments, it's hard to draw a convincing conclusion.

3. The presentation can be improved. First of all, I find the reliance on the computational graph terminology to be quite unnecessary, and it adds difficulty for the precise understanding of the details. Second, the experiments section can be much better organized, with tabulated results and clear comparisons to baselines.

**Summary Of The Paper:**

This paper proposes an improved version of batch normalization that's intended to work well on small training batch sizes. The technique is similar to batch renormalization, but adopts several techniques to manipulate the backprop procedure. Experiments are are conducted primarily on the Inception V3 network, with comparisons to BN and batch renorm.

**Summary Of The Review:**

Overall, I find that this paper does not meet the bar for ICLR. I suggest the authors work on improving the methodology, adding more evaluations and improving the presentation.

---

### Decision · Program_Chairs · 2022-01-20

**Decision:**

Reject

**Comment:**

The paper studies the introduction of a variant of batch normalization (BN) to train deep neural network. The underlying idea is a two-step approach for per-sample based normalization, relying on augmenting the computational graph to handle "several samples" nodes.

The reviewers have mentioned that the idea of altering the computational graph is interesting and potentially novel.
Yet, the numerical experiments were not enough precise or solid to back up the claims by the authors, that their proposed BN alternative is of practical interest.
It was also raised that the paper lacks theoretical supports: no formal analysis, most explanations are ad hoc, etc.